# Characterization of the Rat Osteosarcoma Cell Line UMR-106 by Long-Read Technologies Identifies a Large Block of Amplified Genes Associated with Human Disease

**DOI:** 10.3390/genes15101254

**Published:** 2024-09-26

**Authors:** Alan F. Scott, David W. Mohr, William A. Littrell, Reshma Babu, Michelle Kokosinski, Victoria Stinnett, Janvi Madhiwala, John Anderson, Ying S. Zou, Kathleen L. Gabrielson

**Affiliations:** 1Genetic Resources Core Facility, Department of Genetic Medicine, Johns Hopkins University School of Medicine, 600 N. Wolfe St., 1034 Blalock, Baltimore, MD 21287, USA; dwmohr@jhmi.edu (D.W.M.);; 2Department of Genetic Medicine, Johns Hopkins Genomics, Johns Hopkins University School of Medicine, 1812 Ashland Ave., Suite 200, Baltimore, MD 21205, USA; 3Cytogenetic Laboratory, Department of Pathology, Johns Hopkins Genomics, Johns Hopkins University School of Medicine, 600 N. Wolfe St., Halsted 281, Baltimore, MD 21287, USAyzou19@jhmi.edu (Y.S.Z.); 4Department of Molecular and Comparative Pathobiology, Johns Hopkins University School of Medicine, 733 N Broadway, Baltimore, MD 21205, USA; 5Sidney Kimmel Comprehensive Cancer Center, Johns Hopkins University, Baltimore, MD 21205, USA

**Keywords:** optical mapping, nanopore sequencing, osteosarcoma, Myc amplification

## Abstract

Background/Objectives: The rat osteosarcoma cell line UMR-106 is widely used for the study of bone cancer biology but it has not been well characterized with modern genomic methods. Methods: To better understand the biology of UMR-106 cells we used a combination of optical genome mapping (OGM), long-read sequencing nanopore sequencing and RNA sequencing.The UMR-106 genome was compared to a strain-matched Sprague-Dawley rat for variants associated with human osteosarcoma while expression data were contrasted with a public osteoblast dataset. Results: Using the COSMIC database to identify the most affected genes in human osteosarcomas we found somatic mutations in Tp53 and H3f3a. OGM identified a relatively small number of differences between the cell line and a strain-matched control animal but did detect a ~45 Mb block of amplification that included Myc on chromosome 7 which was confirmed by long-read sequencing. The amplified region showed several blocks of non-contiguous rearranged sequence implying complex rearrangements during their formation and included 14 genes reported as biomarkers in human osteosarcoma, many of which also showed increased transcription. A comparison of 5mC methylation from the nanopore reads of tumor and control samples identified genes with distinct differences including the OS marker Cdkn2a. Conclusions: This dataset illustrates the value of long DNA methods for the characterization of cell lines and how inter-species analysis can inform us about the genetic nature underlying mutations that underpin specific tumor types. The data should be a valuable resource for investigators studying osteosarcoma, in general, and specifically the UMR-106 model.

## 1. Introduction

Cancers can result from a variety of genomic errors, including chromosomal events (translocations, amplifications, deletions, etc.), somatic mutations, and epigenetic changes, all of which may be reflected in altered gene expression. Osteosarcoma (OS) is a rare, largely pediatric cancer of the bone [1,2,3]. Cell lines made from OS and other cancers are often created to better understand and model disease. During passaging, genetic changes can accumulate [4], and it is essential to characterize these to understand how faithful the model is to the disease of interest and to understand the experimental behavior of such cell lines. UMR-106 is widely used in studies of OS [5,6,7,8,9,10,11,12,13,14,15,16] and serves as a model of pediatric cancer, such as the characterization of the tumor microenvironment following implantation into rat tibias [14]. A thorough characterization of UMR-016 is necessary to determine how similar the line is to human OS and whether we are to use this model to explore potential targets for drug therapy.

At present, no single technology permits a thorough characterization of cancer at all genomic scales. However, newer long DNA techniques [17,18] suggest that they may offer improved insights into the tumor genome. Here, we chose to evaluate two long DNA technologies, optical genome mapping (OGM), as developed by Bionano Genomics, and nanopore sequencing, as implemented by Oxford Nanopore Technologies (ONT), since both assess the genome at larger scales than typical short-read methods. Although OGM has been used to study human sarcomas [19], it has not yet been applied to characterizing osteosarcoma genomes. Likewise, long nanopore reads have been shown to be a valuable tool in cancer research (e.g., [20,21]), but only a few studies have compared OGM to ONT [22]. In this study, we evaluated how well both long-read technologies compare and what new insights they provide over traditional methods. We also provide the community with the resources produced for subsequent studies.

## 2. Methods

### 2.1. UMR-106 Cell Culture

Rat osteosarcoma UMR-106 cells (CRL-1661) were purchased from ATCC and expanded for three passages. Cells were grown in DMEM media supplemented with 10% (*v*/*v*) FBS (Fetal Bovine Serum), penicillin (10 U/mL), and streptomycin (10 U/mL) at 37 °C in a humidified 5% CO_2_ atmosphere. Cells used for characterization were within passages 3–6.

### 2.2. Bionano OGM Methods

DNA from UMR-106 cells and from WBCs of the control rat were isolated. Optical Genome Mapping (OGM) methods included ultrahigh-molecular-weight (UHMW) genomic DNA (gDNA) isolation from UMR-106 cells and from WBCs of the control animal, isolated via the blood and cell culture DNA Isolation Kit according to the manufacturer’s instructions (Bionano Genomics, San Diego, CA, USA). Briefly, cells were treated with lysis-and-binding buffer (LBB) to release gDNA, which was bound to a Nanobind disk before it was washed and resuspended in the elution buffer. The Direct Label and Stain (DLS) DNA Labeling Kit (Bionano Genomics, San Diego, CA, USA) was used to label UHMW gDNA molecules. A total of 750 ng of gDNA was labeled with Direct Label Enzyme (DLE-1) and DL-green fluorophores. The G3.3 chips were used, and samples were processed on a Bionano Saphyr instrument (San Diego, CA, USA). OGM analysis was performed using the Rare Variant Analysis (RVA) pipelines [23] and the Bionano Access^TM^ software v1.8.2 [24]. CNVs and SVs were manually determined using 1 Mb as the size cutoff for SVs and 5 Mb as the size cutoff for CNVs. OGM was performed on UMR-106 cells and WBC DNA from a Sprague–Dawley (SD) rat control. In the absence of an SD reference sequence, maps were compared to the rat genome reference rn7.2.

### 2.3. Nanopore Sequencing (ONT)

HMW DNA was isolated from freshly expanded rat osteosarcoma UMR-106 cells using the NEB Monarch kit (NEB #T3060) (New England Biolabs Inc., Ipswich, MA, USA), and an ultralong ONT library was prepared using the SQK-ULK114 kit (Oxford Nanopore Technologies, Oxford, UK). A Qiagen DNAeasy Blood and Tissue kit (#69506) (QIAGEN Sciences Inc., Germantown, MD, USA) was used to isolate DNA from the blood of a female Sprague–Dawley rat, and a standard ONT library was made (SQK-LSK114). The ULK library was made on the cell line in anticipation of likely structural variants. Both libraries were run on 10.4.1 PromethION flow cells with triple loading. The UMR-106 produced 45.3 Gb of aligned bases (to the rat rn7.2 reference), with an N50 of 78 kb. Data were aligned to reference using minimap2 [25]. Methylation calls were generated using modbam2bed 0.9.4 [26]. Modbamtools 0.4.8 was also used to plot methylation data. Summary statistics for the runs were generated with Nanoplot v1.19.0 [27].

### 2.4. RNASeq Methods

Isolated RNA from UMR-106 was poly-A selected using the NEBNExt Poly(A) isolation module (NEB #E7490), and libraries were made using NEBNext Ultra II RNA Library Prep Kit for Illumina (Cat# E7775) (New England Biolabs Inc., Ipswich, MA, USA). Sequencing was performed on a NovaSeq 6000 SP 2 × 50 bp run. FASTQ data were aligned to reference using the STAR aligner 2.7.10b [28]. Control osteoblast data were from SRR16368266. A total of 800 million reads were obtained from UMR-106 and compared to the 45 million available from the osteoblast control. UMR-106 data were downsampled to match read length and approximate depth using samtools 1.17 [29]. RNAseq reads were normalized between osteoblast and UMR-106 reads using GAPDH. Replicates were not available.

### 2.5. Mutation Identification

We focused on genes identified in the COSMIC database ([30]; https://cancer.sanger.ac.uk/cosmic; accessed on 10 May 2023). A detailed analysis was limited to the top twenty genes most commonly associated with human osteosarcoma. We also submitted variant call files for the UMR and control rat genomes to the Variant Effect Predictor ([31]; https://useast.ensembl.org/info/docs/tools/vep/index.html; accessed on 10 May 2023). We searched the output files for variants with “high” predicted designations in the tumor cells, which were absent in the control genome.

## 3. Results

### 3.1. Optical Genome Mapping

Optical mapping identified differences in UMR-106 relative to a healthy Sprague–Dawley (SD) strain sex-matched control animal principally in large blocks of amplified DNA including Myc. A tumor/normal variant analysis was performed using the Bionano Access^TM^ software and the NCBI rn7.2 rat genome reference sequence. Figure 1 shows differences in copy number across chromosome 7.

### 3.2. Correlation between Optical Mapping and Nanopore Sequence Depth

The region of significantly increased copy number identified by OGM was compared to ONT read depth. We normalized the nanopore sequence reads for UMR-106 and the SD control and compared read depth for the region showing relative amplification between UMR-106 and the control. Figure 2 shows that both optical mapping and nanopore sequence depth identify significantly similar amplification across the regions identified above.

### 3.3. Osteosarcoma Mutation Analysis 

ONT sequence was analyzed for candidate genes from human studies of osteosarcoma reported in COSMIC, the Catalogue of Somatic Mutations in Cancer database ([30]; https://cancer.sanger.ac.uk/cosmic; accessed on 12 October 2023). Because a Sprague–Dawley rat genome reference was not available at the time of analysis, and because some variants might reflect strain differences, we chose to sequence an SD control for comparison. To increase the likelihood of identifying structural variants (SVs) relevant to tumorgenicity, we used an ultralong-read library for the UMR-106 cells (N50 ~78 kb). A standard ONT library (~20 kb N50) was generated for the control animal.

Using the top 20 mutated genes in human osteosarcoma listed in COSMIC, we did a manual review of UMR-106 versus the SD control using the Integrated Genomics Viewer (IGV) ver 2.16.2 [32]. We also reviewed other reported cancer genes for copy number, somatic mutations, and changes in 5mC methylation. In the Tp53 gene, UMR-106 DNA had a Leu-to-Phe change at codon 192 (chr10:54,308,638 C>T; VAF 1.0), which has been reported as likely pathogenic. This mutation has also been seen in a rat endometrial tumor [33]. In human cancers, the orthologous codon 194 position is reported in ClinVar ([34]; (https://www.ncbi.nlm.nih.gov/clinvar/RCV000417813.1/; accessed on 12 October 2023) and in chronic lymphocytic leukemia [35]. Manual inspection of data generated with the Variant Effect Predictor [31] tool (VEP release 110) comparing UMR-106 to the control genome did not identify any obvious pathogenic single nucleotide or small indel mutations in other cancer-related genes. However, a large deletion was detected in one-fourth of UMR-106 reads for H3f3a, a gene listed among the top twenty COSMIC loci associated with human OS. As shown in Figure 3, we observed a deletion of 10,995 bases in the affected reads. The figure highlights the utility of long reads in the identification of structural variants that might otherwise be missed.

### 3.4. Expression Data

Short-read RNA sequencing was done on poly-A RNA from the UMR-106 cell line and compared to a public dataset from SD rat osteoblasts ([36]; run SRR16368266). We downsampled and normalized the UMR reads relative to GAPDH in the osteoblast data. Relative expression was used to compare genes in UMR-106 from the amplified region identified by OGM and ONT. As shown in Appendix A, we found that most genes in the region showed significantly increased expression relative to the osteoblast data.

### 3.5. Methylation

Because 5mC status can be directly measured from ONT sequencing, we explored the UMR-106 data for genes that differed in epigenetic patterns from the SD control. The human tumor suppressor CDKN2A has been reported to be more methylated in OS and to be predictive of progression [37,38]. Analysis of the ultralong UMR-106 Cdkn2a data showed hypermethylation at its 3′ end in exons 2 and 3 (Figure 4a). For Lsamp, there was notably less methylation in UMR-106 than in the SD control sample (Figure 4b). Downregulation of Lsamp has been implicated in lung cancer progression and poor prognosis [39,40].

### 3.6. Chromosome 7 Amplified Gene Region

Based on the OGM data showing amplification of the region around Myc, we further explored this part of chromosome 7 with both ONT and RNA sequencing. Genes were examined for normalized read depth between UMR-106 and the SD control and for expression for UMR-106 versus an osteoblast dataset available at NCBI [36]. To reduce skewness [41], log(10) values were calculated for the ratios of tumor to control for OGM molecule depth, ONT sequence depth, and TPM relative expression.

For Myc, we observed an approximately 7×-fold increase in sequence copy number and a 3.5× increase in the number of RNA reads (Appendix A). For the adjacent Ndrg1 gene, a reported biomarker for human osteosarcoma [42,43], we observed about a 7.5-fold increase in copy number and a 3.2 increase in transcription. Other genes reported as markers of human OS are listed in Table 1. Of note, a large number of genes in this block of about 45 Mb have been reported as markers in OS or in other cancers. Myc amplification has been well known in cancer for over 40 years and is often associated with chromothripsis and with generally poorer prognosis [44]. While chromosome breakage and the co-amplification of other genes near Myc have been reported, to our knowledge, there have not been comprehensive studies of the sub-chromosomal amplicons due to the absence of methods to characterize long DNA molecules other than cytogenetically. We looked at additional genes associated with OS in humans and dogs not within the amplified block, including Pten [45], Magi2 [45], Rb1 [45], Dst [45], Dlg2 [45], Dmd [45], and Wwox [46]. None showed evidence of somatic mutations, copy number, methylation, or significant expression differences relative to control data.

**Table 1 genes-15-01254-t001:** Genes in or near the chromosome 7 region identified by OGM (Figure 1). The log(10) ratios of normalized UMR to SD control are calculated from Appendix A. RNA expression was normalized for UMR and SSR osteoblast data relative to GAPDH. The log(10) Tumor/Normal (T:N) ratios for OGM and ONT were compared using a Pearson correlation test and gave an r = 0.957 and a *p*-value of 1.38 × 10^−18^ (see Figure 2). The log(10) of the ONT read depth to the expression TPM ratio gave an r = 0.411 and a *p*-value of 0.018, excluding the low expressing Csmd3, Gsdmc, Kcnq3, Col21a1, and Mfng genes. Genes reported as osteosarcoma markers in the literature (OS marker) or otherwise associated with cancer are indicated. TAF2 has been reported to be amplified in breast cancer and CSMD3 as mutated in Esophageal Squamous Cell Carcinoma. N.B. Gene symbols are capitalized when referring to human genes and with a leading capital letter followed by lowercase letters when referring to rat genes [47,48].

Gene	Chr	Start	End	OGM T:N	ONT T:N	TPM T:N	Comments, References
				Ratio log(10)	Ratio log(10)	Ratio log(10)	
*Mdm2*	7	53290660	53315205	0.167	0.037	−0.209	OS amp; [49]
*Mdm1*	7	53729603	53766034	0.207	0.064	0.419	
*Oxr1*	7	72528750	72965666	0.167	0.127	−0.109	
*Angpt1*	7	73528345	73783953	0.196	0.170	−0.703	OS marker; [50]
*Csmd3*	7	78747322	80066466	0.500	0.473		mut in ESCC; [51]
*Trps1*	7	81916668	82142733	0.541	0.544	0.374	OS marker; [52]
*Eif3h*	7	83091037	83174451	0.753	0.737	0.887	OS marker; [53]
*Taf2*	7	86422613	86479616	0.706	0.639	0.952	BRC amp; [54]
*Deptor*	7	86514859	86668817	0.721	0.604	−0.707	OS marker; [55]
*Has2*	7	88113326	88139337	0.649	0.515	−1.152	OS marker; [56]
*Zhx2*	7	89226358	89374266	0.744	0.719	−0.222	[57]
*Fam91a1*	7	89969558	90007546	0.758	0.727	0.796	OS marker: [58]
*Tmem65*	7	90336997	90378930	0.751	0.776	0.838	OS marker; [59]
*Rnf139*	7	90439726	90450911	0.793	0.767	0.928	OS marker; [60]
*Myc*	7	93593705	93598633	0.759	0.859	0.544	OS marker; [60]
*Gsdmc*	7	95594015	95606106	0.412	0.531		[61]
*Cyrib*	7	95633876	95760588	0.389	0.511	0.719	[62]
*Asap1*	7	95786130	96093111	0.562	0.512	0.580	[63]
*Adcy8*	7	96417310	96665911	0.457	0.508	0.699	[64]
*Efr3a*	7	97552677	97633369	0.696	0.695	0.755	[65]
*Kcnq3*	7	97730219	98025652	0.680	0.744		
*Phf20l1*	7	98330580	98396526	0.552	0.496	0.089	[66]
*Ccn4*	7	98645238	98677253	0.558	0.464	0.748	OS marker; [67]
*Ndrg1*	7	98684487	98725869	0.708	0.873	0.508	OS marker; [42]
*St3gal1*	7	98845270	98913409	0.922	0.882	0.236	OS marker; [68]
*Zfat*	7	99886954	100054288	0.624	0.641	0.637	[69]
*Khdrbs3*	7	100837707	100995644	0.636	0.604	1.655	[70]
*Col22a1*	7	103730939	103968452	0.021	0.037		OS marker; [71]
*Trappc9*	7	104521593	104998352	0.068	−0.092	−0.073	[72]
*Chrac1*	7	105013047	105016435	0.095	0.185	0.091	[73]
*Mfng*	7	110310810	110328653	0.096	0.111		[74]

The fine structure of the chromosome 7 amplified region is complex, as suggested by both ONT sequence and OGM molecule depth across the region. Figure 5 shows a plot for the ONT ultralong reads along with a detailed view of one region to illustrate local complexity and sequence scrambling. Examination of the sequence along the amplified region found many instances of abrupt transitions in UMR-106 relative to the reference but not in the control animal. Because many of the long UMR-106 ONT reads extended beyond the highlighted block, we could confidently map the locations of secondary alignments against the reference. For example, the ~15 kb block near 98.3 Mb has reads that both align well to the reference and also near 98.7 and 57.4 Mb (Figure 5). Similar instances can be seen throughout the amplified region, implying a history of rearrangements and other events in the formation of UMR-106.

## 4. Discussion

As OGM software, by default, ignores molecules smaller than 150 kb, it is able to identify large-scale structural variants several orders of magnitude larger than what can be seen with short-read sequencing. The resolution of OGM is reported to be from 5 to 10 Mb down to a few kb [75]. Since nanopore sequencing can now routinely produce reads with N50s of 20–100 kb or longer, it can effectively bridge the scale afforded by OGM for analysis of smaller SVs and SNVs. Pei et al. [22] recently compared both long DNA methods and concluded that the precision of OGM was very high and that ONT sequencing outperformed short reads for SV detection. In this manuscript, we have combined both long DNA approaches to evaluate their utility in characterizing a widely used osteosarcoma cell line. However, because we anticipated many cancer-related structural anomalies in the UMR-106 cells, we used ONT transposase-based ultralong reads on freshly grown cells isolated to minimize DNA fragmentation. The sequencing produced reads with an N50 of ~78 kb, with many exceeding 100 kb and a few outliers over 200 kb.

Both OGM and ONT identified excess molecules and sequence depth, respectively, corresponding to large portions of chromosome 7 that included the Myc gene, a likely driver of oncogenicity for UMR-106. The correspondence between OGM and ONT depth measurements agreed well (Figure 2). When we explored the region in greater depth using the ultralong reads, we concluded that the fine structure of the region was more complicated than simple alignments show. In Figure 5, the sequence depth can be seen to vary across a large portion of chromosome 7, and some regions showed what initially appeared to be local amplification. However, because of the ultralong reads, we were able to identify that many of these had secondary alignments elsewhere on chromosome 7, often tens of Mb away. The example shown (Figure 5) from a region near 98.3 Mb includes both reads that are in agreement with the reference and others that were non-contiguous, with long secondary alignments mapping near 98.7 and 57.4 Mb. This is best explained as sequence scrambling followed by amplification. We conclude that the amplification of genes along chromosome 7 is not the result of a simple duplication or multiplication event but more likely from a series of events involving complex rearrangements and, possibly, the formation of abnormal chromosomes such as rings, double-minutes, or others. Other than chromosome 7, both OGM and nanopore sequencing identified a relatively small number of genomic changes, suggesting that the UMR-106 cell line has been relatively stable despite its many years in culture, although this requires additional analysis.

As noted above, there was excellent correspondence between OGM and ultralong ONT sequence, and each has advantages and disadvantages, depending on the goals of a particular study. Ultralong ONT reads that approach OGM in length provide base level sequence and 5mC status and can detect CNVs and indels below the resolution of OGM. However, OGM can provide a higher-level view of the genome, which is more difficult to characterize with sequence data alone. When used in combination, they provide complementary data sets. Additionally, ONT sequencing has a clear advantage over optical mapping when exploring genomes in detail since it allows single-base resolution. This is illustrated by the detection of the 11 kb deletion in H3f3a (Figure 3), where we found reads that spanned the deletion and were able to precisely map its boundaries. Sequencing also allowed mutation detection and identified the Tp53 variant. However, the relative stability of UMR-106 was confirmed by manual inspection of dozens of known oncogenes by comparing variant call files between the cell line and the control animal. Alignment of both the SD control and UMR-106 also identified shared variants not seen in the reference and which are likely strain-specific. The ability to call methylated bases without additional sample manipulation was an ancillary benefit of ONT sequencing. The identification of marked differences in 5mC methylation for Cdkn2a and Lsamp in the tumor was consistent with reports of their role in cancer. Going forward, it will be interesting to reanalyze UMR-106, especially with methylation analysis, as characteristics of the cell line change with passage.

An unexpected observation of this study was how many of the genes in the amplified region are associated with literature implicating their role in this cancer type (Table 1 and Appendix A). Oncogene amplification is a well-known phenomenon in cancers, especially for Myc [44], and genes flanking Myc can be co-amplified. Parris et al. [54] noted that TAF2, NDRG1, and TRPS1 were among the genes co-amplified in breast cancer along with MYC on human chromosome 8q. The fact that 14 of the genes we identified in the amplified region are already described as OS markers suggests that many of these genes may cooperate in influencing OS progression and could be targets for intervention. Further studies comparing osteosarcomas with Myc amplification might be useful in defining which adjacent genes are most important in progression. Although the RNAseq data from UMR-106 is limited to a single experiment and the publicly available osteoblast data set is fairly small, we did detect a trend for genes in the amplified region as having increased expression (Appendix A). However, this observation will require confirmation with further studies, especially with additional osteoblast sequences.

This report is a non-exhaustive overview of a single rat osteosarcoma cell line that has been used for over 50 years. The study’s purpose was to explore the advantages of long DNA methods to more rapidly, thoroughly, and rigorously characterize this cell line. Because we only studied a single osteosarcoma line from one species, we cannot unequivocally conclude which changes are most significant in determining the tumor phenotype. However, the commonality of shared genes from human, rat, and canine [45] osteosarcomas and other tumors with Myc amplification suggests shared pathways occur across species. The fact that so many co-amplified genes we observed surrounding Myc are reported as OS markers suggests that they are either commonly amplified passengers along with Myc or have functions that augment Myc. The methods we highlight here are rapid and comprehensive but do require high molecular weight DNA, so they will be limited to fresh samples or established cell lines. We expect that the use of optical mapping and long-read sequencing will simplify the study of amplification and chromothripsis and will provide a deeper understanding of genome architecture in cancer, perhaps providing new strategies for treatment.

## Figures and Tables

**Figure 1 genes-15-01254-f001:**
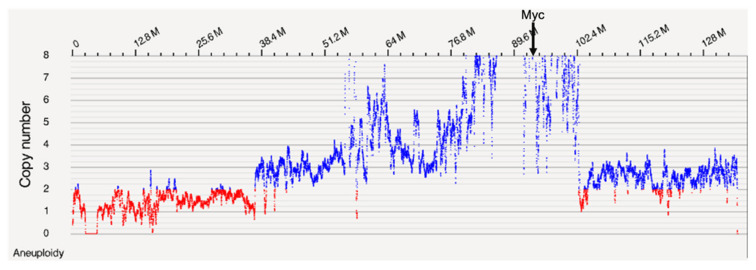
OGM copy number along chromosome 7 with the position of Myc indicated.

**Figure 2 genes-15-01254-f002:**
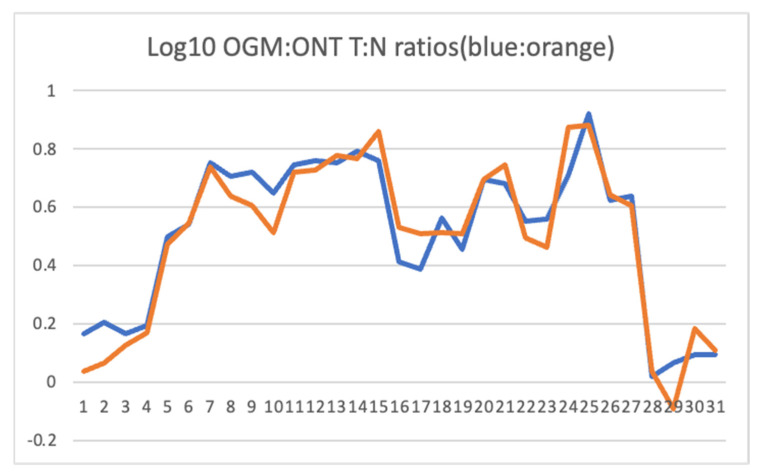
Plot of log(10) values for the ratios of OGM and ONT depth (molecules or sequence reads) of UMR to the control SD animal across genes in the amplified region shown in Figure 1. Data are from Table 1 below.

**Figure 3 genes-15-01254-f003:**
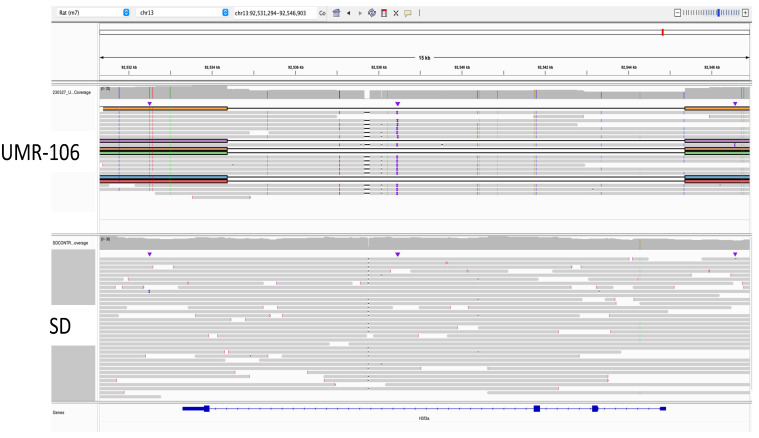
Identification of an H3f3a 11 kb deletion in marked reads from ultralong ONT sequencing. The direction of transcription is from right to left. In the affected reads, exons 1–3 are deleted.

**Figure 4 genes-15-01254-f004:**
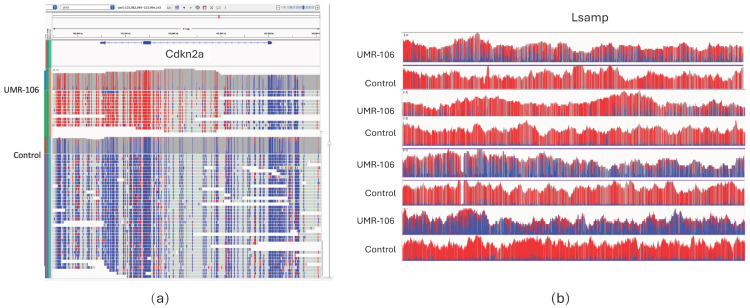
(**a**) IGV display showing hypermethylation (red) at the 3′ end of Cdkn2A, including exons 2 and 3. (**b**) Lsamp methylation plots comparing UMR-106 to SD. The direction of transcription for both genes is right to left. Red represents methylated bases and blue are unmethylated.

**Figure 5 genes-15-01254-f005:**
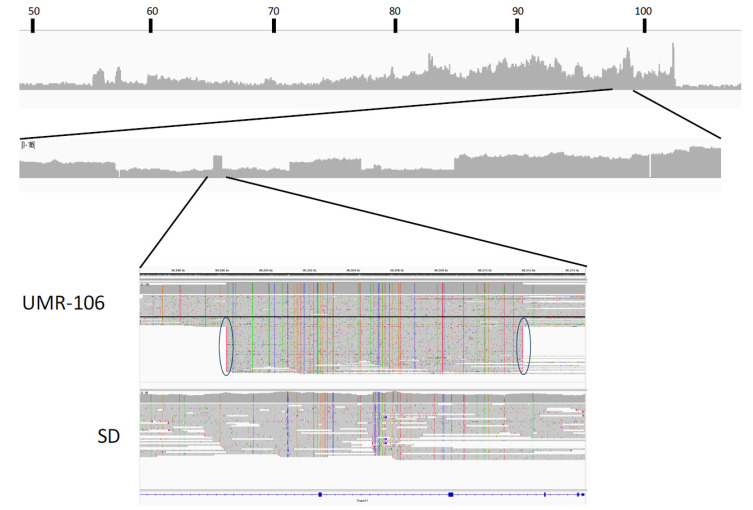
Complexity of the chromosome 7 amplified region as demonstrated by ONT sequence. Overall, ONT sequence depth from ~50 Mb to ~107 Mb along the rat 7.2 reference genome is shown with increased detail from 98 to 99.1 Mb. The bottom IGV image shows alignments at ~98.3 Mb for UMR-106 vs. the SD control. The black-highlighted sequence shows a single read that extends across the block and agrees with the reference, while the ovals highlight reads that have supplementary matches to regions near 98.7 and 57.4 Mb.

## Data Availability

The resources described in this study are available for further exploration by the research community. The data are available at NCBI under project number PRJNA1148449 and include fastq files for the UMR-106 and SD control ONT reads, UMR-106 RNAseq fastq reads, and the OGM optical map files.

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
