# Peer review of "Characterization of the Rat Osteosarcoma Cell Line UMR-106 by Long-Read Technologies Identifies a Large Block of Amplified Genes Associated with Human Disease"

_genes, 2024, doi:10.3390/genes15101254_

Round 1
Reviewer 1 Report
Comments and Suggestions for Authors
The manuscript (brief report) entitled "Resource: Characterization of the rat osteosarcoma cell line UMR-106 by long read technologies identifies a large block of amplified genes associated with human disease" by Alan F Scott et al, presents data on the rat osteosarcoma cell line UMR-106 obtained using long read technologies.
The data reports in the manuscript are interesting and will be useful for researcher on osteosarcoma more specifically those working with the UMR-106 cells.
The Reviewer has a few questions about this manuscript to which he would appreciate answers.
First, osteosarcomas are known to have complex genetics with the presence of polyploidy. Do the authors have any data on the karyotypes of UMR-106? According to the data presented, only chromosome 7 has a duplicated region, with the other chromosomes showing point mutations (e.g. Chr10 for TP53) or intra-gene deletions (e.g. Chr13 for H3f3a). Can we conclude that there is no translocation in UMR-106? A few lines of discussion on this subject would be most welcome!
Concerning the duplicated region of chromosome 7, how do the authors explain that some genes are duplicated 7 times, others 3 times, even though they are neighbors? Similarly, in Figure 1, between 51.2M and 64M, there is an unduplicated region! How can this be explained?
The authors write that the MYC gene is duplicated 7 times, but in the table the number of copies is 11.55! Can you explain this?
Finally, the control used is Sprague-Dawley (SD) rat DNA, which is an outbred rat line and therefore highly susceptible to genetic variation over time and generations. UMR-106 was generated several decades ago, so probably from a SD rat "genetically" different from today's SD rats, as confirmed by the fact that injection of UMR-106 cells into non-immunocompromised SD rats does not work today. Can the authors clarify this point in the manuscript?
Author Response
Response: We do not have karyotypes. The amplification is not a single event and is more complicated than it appears when sequences are simply aligned to the reference. We added a new figure (Fig 5), and discuss this in the results and discussion. We show only a single example and many similar events have occurred over the larger amplified region(s). We suspect, but have not proven, that the cell line may have a normal chr 7 and many minichromosomes, perhaps double minutes, that have amplified to different amounts. We added a figure and discussion to this point.
Thank you for spotting the discrepancy in CN in Table S1. There was a transposition error from Excel that has been corrected in the revision.
We agree with the reviewer that genetic variation over time and multiple generations is likely present when comparing the current cell line and the current animal model Sprague Dawley rats that we used. We originally compared the UMR sequence to the NCBI rat (non-SD) reference and found quite a few differences including some in known oncogenes and it was unclear if they were strain differences or errors in the rat reference. Because of that we sequenced the SD control animal and saw that most differences disappeared. Implanting UMR-106 cells into the tibia in young 3-4 week old non-immunocompromised male or female rats does manifest into tibial osteosarcoma with metastasis to the lung (see PMID 33999035) and is being done by KG’s lab. The rat model demonstrates the same metastasis behavior as is seen in human osteosarcoma. But, to reproduce our model, it is very important to inject the cells into the tibial bone marrow metaphysis/diaphysis site for the tumor implantation to be successful.
Reviewer 2 Report
Comments and Suggestions for Authors
The main objective of this paper is to characterize the rat osteosarcoma cell line UMR-106 using long DNA technology. While this study is very interesting, there are several concerns that should be discussed.
1. what are the major genetic variants in the UMR-106 cell line?
Please elaborate on which genes are particularly mutated in the UMR-106 cell line and how these mutations are associated with osteosarcoma.
2. how does optical genome mapping (OGM) compare to nanopore sequencing?
Describe what new findings were obtained when comparing the results of OGM and nanopore sequencing, especially how each technique provided different information.
3. what are the characteristics of the gene expression profile of UMR-106?
Based on your analysis of gene expression data from the UMR-106 cell line, which genes are particularly highly expressed and discuss their biological significance.
4. how do different methylation patterns affect the progression of osteosarcoma?
Please explain specifically how the altered methylation pattern in UMR-106 affects the progression and prognosis of osteosarcoma.
5. what are the limitations of this study and future research directions?
Please point out the limitations in this study and describe future research directions and proposed experimental approaches based on these limitations.
6. how can this research be applied to immunology research? Please discuss with reference to the following paper.
Clinicopathological assessment of cancer/testis antigens NY‑ESO‑1 and MAGE‑A4 in osteosarcoma. Eur J Histochem. 2022;66(3):3377. Published 2022 Jun 23. doi:10.4081/ejh.2022.3377
Author Response
As noted in the manuscript the predominant change is the amplification of Myc and flanking genes associated with OS as well as the point mutation in Tp53. We did not find any other oncogenic changes in the major genes associated with OS on the COSMIC database or by comparing UMR to control VCFs using VEP.
As shown in Fig 2 there is a very good correlation between molecule depth from OGM and sequence reads from ONT. We focused on the ultralong ONT reads since they are a relatively new technique and come closest to matching OGM in length. Along with single-base resolution they also provided methylation status. We’ve added text to the discussion to try to clarify this.
All the RNAseq data has been submitted to the NCBI SRA. The caveat with this part of the study is that there was relatively little data from the SD control osteoblast study and robust expression studies require multiple replicates. For that reason we only focused on expression of genes from the amplified region with the premise that they might be upregulated along with copy number. The data are shown in Table S1 (now corrected) found that several genes from the amplified region of chr 7 are overexpressed compared to the osteoblast control data. Those genes are also shown in Table 1 along with references to their association with OS.
We don’t have data relevant to the question of progression. But, by presenting this baseline data we hope that others who use this cell line will be able to follow up by noting phenotype changes, such as increased growth or metastatic characteristics in their studies. We added a comment about that in the discussion.
Limitations related to OS is that this is a single cell line and is not necessarily applicable to all cases of osteosarcoma. Also, as noted, there is relatively little rat osteoblast RNAseq so making inferences about expression differences is limited and focused on the amplified genes. No additional studies related to this manuscript are planned although we hope that others will find this resource helpful. A primary goal of the manuscript is to show that these techniques, esp. ONT sequencing, are fairly easy and inexpensive and that all widely used cell lines should be characterized just as any other laboratory reagent both to confirm their authenticity and to identify genomic changes that may occur during passage.
Thank you for the interesting reference. There were no SNVs seen in MAGEA4 between UMR and control. NY-ESO-1 (CTAG1B) is not annotated in the rat reference genome so it may be human specific. Ctag2, also on the X, did not show any SNVs that were different between UMR and control. As with other tumors, identifying UMR neoantigens could be useful for activating T-cells as a strategy for treatment but this is outside our scope. Hopefully, making these data available will encourage other labs to look for possible markers in cell surface proteins to test that idea.
Round 2
Reviewer 2 Report
Comments and Suggestions for Authors
The authors replied well, so the manuscript is suitable foe publication.